# An Exploration of Small Molecules That Bind Human Single-Stranded DNA Binding Protein 1

**DOI:** 10.3390/biology12111405

**Published:** 2023-11-06

**Authors:** Zachariah P. Schuurs, Alexander P. Martyn, Carl P. Soltau, Sam Beard, Esha T. Shah, Mark N. Adams, Laura V. Croft, Kenneth J. O’Byrne, Derek J. Richard, Neha S. Gandhi

**Affiliations:** 1Centre for Genomics and Personalised Health, School of Chemistry and Physics, Queensland University of Technology (QUT), Brisbane, QLD 4000, Australia; zachariah.schuurs@hdr.qut.edu.au (Z.P.S.); alexander.martyn@qut.edu.au (A.P.M.); 2Cancer and Ageing Research Program, Centre for Genomics and Personalised Health, Queensland University of Technology (QUT), Translational Research Institute (TRI), Woolloongabba, QLD 4102, Australia; sam.beard1@gmail.com (S.B.); mn.adams@qut.edu.au (M.N.A.); laura.croft@qut.edu.au (L.V.C.); k.obyrne@qut.edu.au (K.J.O.); derek.richard@qut.edu.au (D.J.R.); 3School of Chemistry and Physics, Centre for Materials Science, Queensland University of Technology, Brisbane, QLD 4000, Australia; c.soltau@qut.edu.au; 4School of Biomedical Sciences, Faculty of Health, Queensland University of Technology, Translational Research Institute, Woolloongabba, QLD 4102, Australia; 5Cancer Services, Princess Alexandra Hospital—Metro South Health, Woolloongabba, QLD 4102, Australia; 6Department of Computer Science and Engineering, Manipal Institute of Technology, Manipal Academy of Higher Education, Manipal 576104, India

**Keywords:** hSSB1, oligonucleotide/oligosaccharide binding-fold, DNA repair, structure-based drug design, ssDNA interactions

## Abstract

**Simple Summary:**

Innovative approaches are required to combat the complexity and adaptability of cancerous cells. Proteins that bind to DNA play an important role in the ability of cancerous cells to survive traditional treatments. This study explores small molecules that bind to one specific protein in these mechanisms—human single-stranded DNA binding protein 1. Using complementary computational and experimental approaches, we discovered three small molecules that appear to prevent the protein from binding to DNA. The computational tools suggest how the compounds bind to human single-stranded DNA binding protein 1, and cellular studies indicate that the molecules may interfere with the cell’s ability to repair DNA at certain concentrations. Further work is necessary to understand how these compounds interact with cells, and to develop them into selective hSSB1 inhibitors.

**Abstract:**

Human single-stranded DNA binding protein 1 (hSSB1) is critical to preserving genome stability, interacting with single-stranded DNA (ssDNA) through an oligonucleotide/oligosaccharide binding-fold. The depletion of hSSB1 in cell-line models leads to aberrant DNA repair and increased sensitivity to irradiation. hSSB1 is over-expressed in several types of cancers, suggesting that hSSB1 could be a novel therapeutic target in malignant disease. hSSB1 binding studies have focused on DNA; however, despite the availability of 3D structures, small molecules targeting hSSB1 have not been explored. Quinoline derivatives targeting hSSB1 were designed through a virtual fragment-based screening process, synthesizing them using AlphaLISA and EMSA to determine their affinity for hSSB1. In parallel, we further screened a structurally diverse compound library against hSSB1 using the same biochemical assays. Three compounds with nanomolar affinity for hSSB1 were identified, exhibiting cytotoxicity in an osteosarcoma cell line. To our knowledge, this is the first study to identify small molecules that modulate hSSB1 activity. Molecular dynamics simulations indicated that three of the compounds that were tested bound to the ssDNA-binding site of hSSB1, providing a framework for the further elucidation of inhibition mechanisms. These data suggest that small molecules can disrupt the interaction between hSSB1 and ssDNA, and may also affect the ability of cells to repair DNA damage. This test study of small molecules holds the potential to provide insights into fundamental biochemical questions regarding the OB-fold.

## 1. Introduction

Cells have evolved a complex network of pathways to maintain a careful balance of genomic stability. Continuous exposure to exogenous and endogenous damaging agents introduces damage to cellular biomolecules, including proteins, lipids, and DNA [1]. Anti-cancer drugs and ionizing radiation (IR) are two common exogenous sources that can lead to DNA damage. DNA damage includes mismatches, single-strand breaks, double-strand breaks (DSBs), crosslinks between strands, and base excision repair (BER), where chemical modifications to bases and sugars are removed [2]. DNA damage and repair (DDR) pathways repair mutations, but the dysregulation of the DDR pathway is associated with cancer, regularly introducing mutations that induce resistance to cancer treatments [3,4].

In human cells, human single-stranded DNA binding protein 1 (hSSB1) (NABP2/OBFC2A) is an essential component of the DDR network [5,6,7]. This single-stranded DNA binding (SSB) protein is an oligonucleotide/oligosaccharide-binding-fold protein [8,9]. It serves multiple purposes in DDR—the initiation of DSB repair through homologous recombination [10,11], the removal of 8-oxoguanine in the BER pathway [6], and the repair of stalled replication forks [10]. Classical cancer treatments induce DNA damage and drive apoptosis. Cancer cells display high rates of DNA repair, which counteracts these therapies and often leads to developing resistance [12,13]. hSSB1 plays a core role in DDR pathways and was recently suggested to modulate cellular responses to androgen and DNA damage in solid malignancies [14]. Accordingly, designing hSSB1 inhibitors might make anti-cancer treatments possible [15].

A limited number of hSSB1 crystal structures have been elucidated as an oligomerized dimer or within the SOSS1 complex, bound to the partner protein integrator complex subunit 3 (INTS3) and C9ORF80 [9]. Those with a ligand in the single-stranded DNA (ssDNA) binding site have been co-crystallized with a chain of deoxythymidine ssDNA (poly-T), showing the interactions between ssDNA and hSSB1 (Figure 1). The residues interacting with hSSB1 slightly differ between the NMR and X-ray crystal structures [15,16]. In the solution structure, ssDNA is modulated by the aromatic residues Trp-55, Tyr-74, Phe-78, and Tyr-85, while in the crystal structure, only Trp-55 and Phe-78 are involved [15]. With an understanding of how hSSB1 binds ssDNA, we developed a structure–activity relationship (SAR) to determine potential protein inhibitors. It has been suggested that SUMOylation inhibitors might inhibit the critical functions of hSSB1 [17], but this has not been tested. Targeting hSSB1 for cancer therapy has the advantage of a limited redundancy system; hSSB2 serves a role similar to hSSB1 and is the only known redundancy system [17,18].

Small molecules with an affinity for hSSB1 have not been published. Small molecule inhibitors have been explored for two other OB-fold proteins, replication protein A (RPA) [19,20] and *E. coli* SSB [21,22]. hSSB1, despite lacking an overall sequence homology with these OB-fold-containing proteins, shares structural similarities in the ssDNA binding interface. Therefore, small molecules with an affinity to other OB-folds were considered as starting scaffolds for the design of hSSB1 inhibitors. Centred on β-strands 2 and 3, the loops between the β-strands define an ssDNA binding cleft that runs perpendicular to the axis of the β-barrel [23]. An inhibitor campaign against the closely related OB-fold RPA has been progressively published over the past eight years [19,20,24]. The resulting lead, TDRL-551 (**1**), binds to the ssDNA binding domains (DBDs) of RPA subunits DBD-A and DBD-B. It demonstrates cytotoxicity when used in conjunction with cisplatin in human lung cancer models [19,20]. The chirality of TDRL-551 is not reported in the literature, which may indicate that both forms are active. The screening of small-molecule antagonists of *E. coli* SSB have mainly targeted protein–protein interactions (PPIs) with no reported structures and IC_50_ values reported as < 40 μM [21], or else only conducting cell-based assays [25]. The only study targeting the ssDNA interaction reported 9-hydroxyphenylfluoron and purpurogallin to be 50% bound at ~25 μM [22].



Cells depleted of hSSB1 cannot repair DNA damage, eventually leading to apoptosis [10]. Here, we present the identification of small molecules that bind to hSSB1 and disrupt the interaction of ssDNA with hSSB1. We chose to approach this biochemical exploration in a bifold manner. The first step was to test TDRL-551 and computationally design a library of similar compounds based on the structural homology of hSSB1 with RPA. The second approach involved screening a physical library of structurally diverse small molecules. Subsequently, we used molecular dynamics (MD) approaches like cosolvent simulations to understand how these compounds bind to hSSB1. The compounds identified in this study are excellent hits, with drug-like properties that are suitable for future characterization in cell-based and in vivo studies.

## 2. Methodology

### 2.1. Structure-Based Design of Compounds

Based on the structural homology of the hSSB1 and RPA OB-fold binding sites, we hypothesized that TDRL-551 and its derivatives would bind to hSSB1. The OB-fold structures of RPA and hSSB1 were isolated using ChimeraX [26,27], before being imported into Flare, where they were prepared by adding missing hydrogens and minimizing the structures. Flare™ (version 6.0, Cresset^®^) [28,29] was used for the initial structure-based design of a series of potential hSSB1 inhibitors. The library of ten compounds from Mishra et al. [19] were docked to the DBD-B of RPA (PDB:1FGU) to validate the docking, and later docked to the hSSB1 (PDB:4OWX) ssDNA binding site. The ligands were imported to Flare™ [28,29] in an SDF file with AutoDetect, and XED force field parameterization was applied. Hydrogens were protonated at pH 7.0, with energy minimization performed and rotatable bonds defined. The grid box was defined by selecting the aromatic residues that interact with poly-dT in the ssDNA binding site of each protein (hSSB1: Trp-55, Tyr-74, Phe-78, Try-85; RPA: Trp-361, Phe-386).

The “accurate but slow” scoring system was used, with a maximum of 10 poses generated for each compound, and the pool size and population size was set to 1. Lead Finder was used to score the predicted binding poses with the rigid protein. The compound with the highest binding score to hSSB1 was then used as the core scaffold in the R group replacement module of Cresset Spark version 10.6 [28] (**2**).

A 5-phenyl-3,4-dihydropyrazole scaffold was used to create a combinatorial library of 64 compounds. Spark performs this by procedurally replacing a specified region R group of the compound with fragments from a library, scoring the new group in the binding region, and ranking them. The top three scoring fragments for each R group and the original three fragments were combined in all possible arrangements using ChemDraw version 20.0 (Table 1). These were minimized, prepared, and subsequently docked to hSSB1 with Flare™ [28,29]. The 15 compounds with the highest LF Rank Score were studied for synthetic feasibility.



**Table 1 biology-12-01405-t001:** Top-scoring R groups from the Spark R group replacement screen.

R1	R2	R3
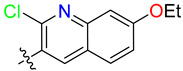	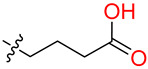	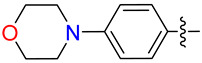
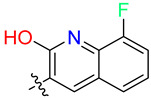	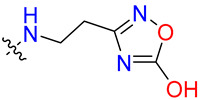	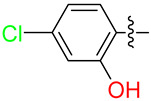
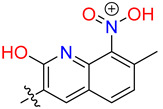	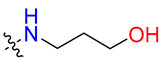	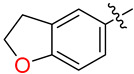
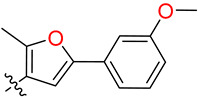	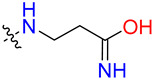	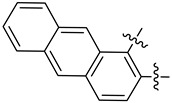

### 2.2. Compound Synthesis

Based on the combinatorial library docking, two compounds were designed and synthesized: DAZLN-55 and DAZLN-56. TDRL-551 [19] was also synthesized as a control molecule. Compound characterization spectra are shown in the Appendix A.

#### 2.2.1. Synthesis of TDRL-551

The synthetic approach used to prepare TDRL-551 was the same as that described by Mishra et al. [19]. The characterization of the molecule matched the published data.

#### 2.2.2. Synthesis of Intermediate



To a stirred suspension of 1-(6-methoxynaphthalen-2-yl)ethan-1-one (500 mg, 2.49 mmol) and 6-methoxy-1-methyl-1*H*-indole-3-carbaldehyde (471 mg, 2.49 mmol) in EtOH (70 mL), 2M NaOH was added (2.60 mL, 5.28 mmol). The solution was heated to reflux for 3 h before being allowed to cool to RT. Upon cooling, a bright yellow precipitate had formed, further encouraged through dilution with excess deionized water. The precipitate was collected and dried via vacuum filtration to give **3** as a bright yellow crystalline solid (739 mg, 80%). ^1^H NMR (DMSO-d_6_, 600MHz): δ 8.76 (d, *J* = 1.8 Hz, 1H), 8.14–8.09 (m, 2H), 8.07 (d, *J* = 8.6 Hz, 1H), 7.99 (d, *J* = 15.4 Hz, 1H), 7.97 (s, 1H), 7.93 (d, *J* = 8.6 Hz, 1H), 7.75 (d, *J* = 15.4 Hz, 1H), 7.43 (d, *J* = 2.5 Hz, 1H), 7.28 (dd, *J* = 8.9, 2.5 Hz, 1H), 7.12 (d, *J* = 2.3 Hz, 1H), 6.92 (dd, *J* = 8.6, 2.3 Hz, 1H), 3.93 (s, 3H), 3.86 (s, 3H), 3.83 (s, 3H) (Appendix A); ^13^C NMR (150 MHz, DMSO-d_6_) δ 188.1, 159.1, 156.6, 139.1, 138.0, 136.6, 135.9, 133.7, 131.2, 129.3, 127.7, 127.1, 125.0, 121.5, 119.5, 119.3, 115.1, 112.1, 111.0, 106.0, 94.3, 55.4, 33.0 (Appendix A). Used without further purification.

#### 2.2.3. Synthesis of DAZLN-51



To a stirred suspension of **3** (300 mg, 0.81 mmol) in EtOH (15 mL), hydrazine hydrate (0.20 mL, 4.04 mmol) was added dropwise before the solution was heated to reflux. After 1 h, the heat was removed, and a white precipitate formed upon cooling. The precipitate was collected and dried via vacuum filtration to obtain compound **4** as a white solid (280 mg, 90%). ^1^H NMR (DMSO-d_6_, 600 MHz): δ 7.95 (dd, *J* = 8.6, 1.7 Hz, 1H), 7.89 (s, 1H), 7.83 (d, *J* = 8.9 Hz, 1H), 7.80 (d, *J* = 8.7 Hz, 1H), 7.43 (d, *J* = 8.6 Hz, 1H), 7.33 (m, 2H), 7.16 (dd, *J* = 8.9, 2.6 Hz, 1H), 7.15 (s, 1H), 6.94 (d, *J* = 2.3 Hz, 1H), 6.64 (dd, *J* = 8.6, 2.2 Hz, 1H), 5.07 (td, *J* = 10.2, 2.8 Hz, 1H), 3.88 (s, 3H), 3.79 (s, 3H), 3.69 (s, 3H), 3.50 (dd, *J* = 16.0, 10.7 Hz, 1H), 3.07 (dd, *J* = 16.0, 9.7 Hz, 1H) (Appendix A). Owing to its stability, **4** was used immediately in the subsequent reaction. Degradation was observed in the immediate ^13^C NMR spectra, so it has not been reported.

#### 2.2.4. Synthesis of DAZLN-55



To a stirred suspension of **4** (300 mg, 0.81 mmol) in CHCl_3_ (10 mL), glutaric anhydride (138 mg, 1.21 mmol) was added. The solution was heated to reflux for 3 h before being allowed to cool to RT, upon which a precipitate had formed. The precipitate was collected and dried via vacuum filtration to obtain compound **5** as an off-white solid (228 mg, 57%). ^1^H NMR (DMSO-d_6_, 600 MHz): δ 12.05 (bs, 1H), 8.13 (d, *J* = 1.7 Hz, 1H), 8.04 (dd, *J* = 8.6, 1.8 Hz, 1H), 7.89 (dd, *J* = 8.8, 4.1 Hz, 2H), 7.39 (d, *J* = 2.6 Hz, 1H), 7.29–7.15 (m, 2H), 7.09 (s, 1H), 6.92 (d, *J* = 2.3 Hz, 1H), 3.90 (m, 4H), 3.76 (s, 3H), 3.67 (s, 3H), 2.75 (td, *J* = 7.5, 3.2 Hz, 2H), 2.25 (m, 2H), 2.14 (t, *J* = 7.4 Hz, 1H), 1.87–1.73 (m, 2H) (Appendix A); ^13^C NMR (DMSO-d_6_, 150 MHz): δ 174.3, 169.3, 158.3, 155.8, 154.7, 137.8, 135.2, 130.5, 128.1, 127.3, 127.2, 126.7, 125.9, 123.7, 119.2, 119.1, 119.0, 114.8, 109.0, 106.3, 93.5, 55.3, 53.1, 40.7, 39.9, 39.8, 39.7, 39.5, 39.4, 39.2, 39.1, 39.1, 33.0, 32.9, 32.8, 32.3, 20.5, 20.0 (Appendix A). Used without further purification.

#### 2.2.5. Synthesis of DAZLN-56



To a stirred suspension of **5** (76 mg, 0.197 mmol) in DCM (15 mL) and ACN (5 mL), DIPEA (135 µL, 0.394 mmol) and HBTU (110 mg, 0.292 mmol) were added. After 20 min, 2-amino-1-morpholinoethanone (34 mg, 0.233 mmol) in DCM (2 mL) was added dropwise and the mixture was stirred overnight. The solution was then diluted with DCM (100 mL) and washed with 1M HCl (30 mL). The organic phase was dried over anhydrous sodium sulphate and concentrated in vacuo. The resulting solid was purified through flash chromatography (CHCl_3_:EtOH) to afford **6** as an off-white solid (80 mg, 66%). M.p = 118–119 °C; ^1^H NMR (CDCl_3_, 600 MHz): δ 8.07 (dd, *J* = 8.6, 1.7 Hz, 1H), 7.90 (d, *J* = 1.5 Hz, 1H), 7.77 (d, *J* = 8.6 Hz, 1H), 7.73 (d, *J* = 9.7 Hz, 1H), 7.25 (d, *J* = 8.8 Hz, 1H), 7.17 (m, 2H), 6.99 (s, 1H), 6.68 (m, 3H), 5.89 (dd, *J* = 11.8, 4.4 Hz, 1H), 4.00 (m, 2H), 3.94 (s, 3H), 3.81 (s, 3H), 3.78 (dd, *J* = 17.4, 11.8 Hz, 1H), 3.67 (s, CH_3_), 3.61 (m, 4H), 3.56 (m, 2H), 3.46 (dd, 17.4, 4.5, 1H), 3.33 (m, 2H), 2.50 (bs, 1H), 2.31 (t, 7.4 Hz, 2H), 2.05 (m, 2H) (Appendix A); ^13^C NMR (CDCl_3_, 150 MHz): δ 173.0, 170.9, 166.8, 158.9, 156.5, 155.0, 138.4, 135.7, 130.1, 128.5, 127.4, 127.1, 127.0, 126.3, 124.2, 119.6, 119.3, 114.7, 109.3, 106.2, 93.4, 66.7, 66.4, 55.8, 55.5, 53.9, 44.9, 42.3, 41.2, 40.8, 35.5, 33.2, 32.9, 21.1 (Appendix A); HRMS (ESI) *m*/*z* calcd. for C_35_H_40_N_5_O_6_^+^: 626.2974 [*M*+H]^+^; found 626.2970 (Appendix A).

### 2.3. Expression and Purification of Recombinant Wild-Type hSSB1

Recombinant hSSB1 with an N-terminal 6x His-tag was cloned into the pET42 vector and transformed into *E. coli* BL21 (DE3*) cells. The cells were grown at 30 °C until an optical density of 0.6 was reached. The expression of hSSB1 was induced in the cells using 0.4 mM ITPG for 12 h at 16 °C. The frozen cell pellets were sonicated in lysis buffer (50 mM Tris-HCl (pH 7.5), 10% sucrose, 10 mM EDTA, 600 mM KCl, 0.01% IGEPAL, CA-630 (Sigma-Aldrich, St Louis, MO, USA)) in the presence of protein inhibitors (aprotinin, chymostatin, leupeptin, and pepstatin, 2 mg/mL each). The cell lysate was centrifuged at 40,000× *g* for 1 h. The supernatant was resolved on a 20 mL SP Sepharose Fast Flow column (GE Healthcare, Chicago, IL, USA) using an NGC FPLC (BioRad, Hercules, CA, USA) with a 5-column volume gradient of 100 to 1000 mM KCl in buffer K (20 mM KH_2_PO_4_ at pH 7.4, 0.5 mM EDTA, 10% glycerol, 0.01% IGEPAL CA-630). Fractions containing hSSB1 were combined and incubated with 10 mM imidazole and Ni-NTA agarose resin (Cytiva, Marlborough, MA, USA) for 2 h at 4 °C, before being washed extensively. Nickel-bound hSSB1 was eluted with buffer K containing 200 mM imidazole and 300 mM KCl, and the fractions containing hSSB1 were combined. A 10 kDa Amicon Ultra centrifugal device (Merck Millipore, Darmstadt, Germany) was used to concentrate the hSSB1 to 250 µL. This was loaded onto a Superdex200 10/300 GL size-exclusion chromatography column (GE Healthcare) and run using K buffer with 300 mM KCl.

### 2.4. AlphaLISA Assay Design

To analyse the binding of compounds to hSSB1 and design a screening assay, we used Perkin Elmer’s AlphaLISA technology [30]. The full assay design is described in the Appendix A. The optimal assay conditions were determined with a series of titration matrices to be 0.5 nM hSSB1, 5 nM In3-PD-biotin, 1 nM anti-hSSB1 antibody, 10 μg/mL SA donor beads, and 5 μg/mL protein G acceptor beads in a 16 μL reaction volume. These were combined in a buffer containing 10 mM Tris (pH 7.5), 100 mM NaCl, 0.1% β-mercaptoethanol, and 0.01% IGEPAL. First, an 8 μL mix of hSSB1, anti-hSSB1 antibody, and acceptor beads was prepared, which was added to the compound of interest and incubated for 1 h. Second, an 8 μL aliquot of a solution containing In3-PD-biotin and SA donor beads was added to the well. This was incubated before imaging with an EnSight Multimode Plate Reader (PerkinElmer, Shelton, CT, USA) at 10, 60, and 120 min. The assay was designed in a competition format so that the Cheng–Prusoff equation was met and the IC50 approximated the K_d_ [31].

### 2.5. Mini Library Compound Library Screen

In parallel to the rational design of small-molecule inhibitors, we acquired access to the Open Innovation Mini Library of small compounds from Merck KGaA, Darmstadt, Germany. This library consisted of 80 compounds at 10 mM in DMSO that had been investigated against other targets, from receptor blockers to protein inhibitors. The goal of screening this library was to test various scaffolds for their affinity to hSSB1.

Using a FlexDrop iQ, 16 nL of each compound was dispensed onto a white low-volume 384-well microplate (PerkinElmer, Shelton, CT, USA) in triplicate at an assay concentration of 10 μM. In3-PS was used as the positive control binding entity at a concentration of 1 μM. The AlphaLISA assay conditions are described in Section 2.4. The hit cut-off used was an average inhibition of 30% across the three repeats. The plate was read using an EnSight Multimode Plate Reader (PerkinElmer, Shelton, CT, USA) at 10, 60, and 120 min.

### 2.6. AlphaLISA Dose–Response Curves

Dose–response curves were obtained for hit compounds and compared with TDRL-551. The same AlphaLISA competition format was used to determine the IC50/K_d_ of compounds binding to hSSB1. Each compound was dissolved in DMSO before being dispensed into a white 384-well microplate (PerkinElmer, Shelton, CT, USA) using a FlexDrop iQ. An immunoprecipitation assay was used to check that the compounds of interest did not interfere with the interaction between hSSB1 and the hSSB1 antibody (Appendix A).

### 2.7. TruHits Counter Assay

The TruHits counter assay was used to identify any false positives in the library screen and designed compounds. This identifies compounds that may interfere with the streptavidin–biotin interaction, colour quenchers, light-scattering compounds, and singlet oxygen quenchers. The compounds that showed some activity in the AlphaLISA assays—DAZLN-55, DAZLN-56, TDRL-551, MS-ML24, MS-ML25, and MS-ML26—were dissolved in DMSO and subsequently dispensed into a white 384-well microplate (PerkinElmer, Shelton, CT, USA) using a FlexDrop iQ. The amount dispensed was between 40 and 680 nL, with a final concentration ranging from 0.125 to 400 μM. DMSO vehicle wells were also used, with volumes ranging from 40 to 680 nL. A solution of TruHit donor beads and streptavidin acceptor beads was prepared in a buffer containing 10 mM Tris (pH 7.5), 100 mM NaCl, 0.1% β-mercaptoethanol, and 0.01% IGEPAL to a bead concentration of 10 μg/mL. The bead solution was incubated for 30 min before adding 16 μL of the solution to the dispensed compound. The plate was covered with optical film and incubated in the dark for 1 h before reading on the EnSight Multimode Plate Reader (PerkinElmer, Shelton, CT, USA).

### 2.8. Electrophoretic Mobility Shift Assays

Electrophoretic mobility shift assays (EMSAs) were used to confirm and validate the binding of molecules that exhibited an affinity to hSSB1 in the AlphaLISA assay. These were performed in 10 μL reactions of 10 mM Tris (pH 7.5), 100 mM NaCl, 0.01% IGEPAL, and 1 mM DTT. Two 5 μL solutions were prepared. The first mix contained hSSB1 with the compound of interest, which were incubated together for 10 min before the addition of the probe molecule, Cy5-labelled In3-PS. The combined solution was incubated for 30 min at 37 °C, before loading dye was added and the samples were loaded into 10% polyacrylamide TBE gels. These gels were run for 1.5 h in a cold room at 80 V before being imaged using a ChemiDoc Imaging System (BioRad, Hercules, CA, USA) and quantified using ImageStudio Lite (Version 5.2) software.

The optimal conditions for the EMSAs were established using a titration matrix of hSSB1 and Cy5-labelled In3-PD concentrations. Concentrations of hSSB1 from 1000 to 0.26 nM and Cy5-labelled In3-PD at 1 nM and 5 nM were tested. The final conditions were established as 15 nM hSSB1 and 1.5 nM Cy5-labelled In3-PS, where the In3-PS is 100% bound. The concentrations of the MS-ML compounds of Merck KGaA, Darmstadt, Germany, were tested from 100 to 0.046 μM, and the concentrations of the novel DAZLN compounds were tested from 200 to 0.091 μM.

### 2.9. Cell Culture and Cytotoxicity Assay

The osteosarcoma cell line U2OS was purchased from the American Type Culture Collection (ATCC) and seeded at a density of 500 cells/well in a 384-well plate. They were maintained at 37 °C in a humidified atmosphere containing 5% CO_2_ in Roswell Park Memorial Institute 1650 (RPMI) medium (ThermoFisher Scientific, Inc., Brisbane, Australia) supplemented with 10% (*v*/*v*) heat-inactivated foetal bovine serum (FBS) (ThermoFisher Scientific). A cytotoxicity assay was conducted to test the impact of the three Merck KGaA, Darmstadt, Germany, compounds (MS-ML24, MS-ML25, MS-ML26) on cell viability. Increasing concentrations of each MS-ML compound were added from 30 to 0492 μM in serial dilutions of 2.5 (*n* = 4). The cells were incubated with the compound for 48 h before the CellTitre-Glo (CTG) (Promega) assay was performed as per the manufacturer’s instructions. The plate was read on an EnSight Multimode Plate Reader (PerkinElmer, Shelton, CT, USA).

### 2.10. Cosolvent Molecular Dynamics Simulations

Molecular dynamics simulation methods and analysis using AMBER20 [32] were used to predict how the molecules would bind to hSSB1. These methods are provided in the Appendix A.

## 3. Results

### 3.1. Recognition of Small Molecule-hSSB1 Binding Site

To explore small molecules with an affinity to hSSB1, we first wanted to understand how TDRL-551, a compound developed by Mishra et al. [19], binds to RPA. A comparison between the ssDNA binding site of hSSB1 and RPA is provided in the (Appendix A). It was suggested that TDRL-505, from which TDRL-551 was derived, was specific to eukaryotic OB-fold-ssDNA interactions after it was tested with *E. coli* SSB and protection of telomeres protein 1 (POT1) [24]. In a comparative docking between hSSB1 and the DBD-B from RPA70, we found that compound **2** had the highest Lead Finder (LF) Rank Score to hSSB1 at −11.541, whereas TDRL-551 only scored −9.946.

The LF Rank Score of TDRL-551 was −8.788 when docked to RPA. This initial docking served a dual purpose, to benchmark the scoring function and to develop the structure–activity relationship, which would serve as the starting point for the further design of compounds. TDRL-551 docked to hSSB1 close to the pose published previously [20]. The pose in Figure 2B shows the Trp-361 interacting with the iodobenzene and the Phe-386 interacting with the quinoline group, matching the published interactions. It is possible that different basic residues in the region stabilise the flexible carboxyalkyl chain. Other top-scoring poses of the compound showed the carboxyalkyl chain flipped in the other direction, stabilized by basic residues such as Lys-343. Conserved tryptophan and phenylalanine residues (Appendix A) in each protein strongly stabilize the molecule through π-π interactions with the TDRL-551 aromatic groups (Figure 2A,B), and while there are two tyrosines in the binding pocket of hSSB1, only the Tyr-85 interacts with the compound in this pose. The third aromatic residue in hSSB1 serves to increase the stability of the compound in the pocket. The SAR developed in this section was the basis for the subsequent screening program.

### 3.2. Combinatorial Virtual Library Screen

To develop and subsequently optimize the affinity of compounds based on the TDRL series to hSSB1-a 4-oxo-4-(5-phenyl-3,4-dihydropyrazol-2-yl)butanoic acid scaffold—we ran an R group fragment growth campaign using the Spark program from Cresset [28]. The core scaffold selected was 3-phenyl-4,5-dihydro-1H-pyrazole-1-carbaldehyde—the core of compound **2**. Only the top three fragments from each R group were selected, and they were combined in all possible combinations (Table 1) to a total of 64 compounds. Appendix A shows the 10 compounds with the highest LF Rank Score from docking. Compound **2** has the highest LF Rank Score from the original benchmark and ranked tenth, with a score of −10.464 when docked with the combinatorial compounds. A carboxylic acid chain at the -R2 position was present in most of the top 10 scoring compounds, and similarly, there were quinoline structures in most of the compounds at the -R1 position. Except for the top-scoring compound, there were more H-bond donors and acceptors in these compounds compared to the six acceptors and one donor in TDRL-551.

### 3.3. Synthesis of Compounds

While we created a series of molecules in the rational design, these are not always synthetically tractable. The compounds that scored highest presented synthetic difficulties, and we therefore needed to find bioisosteres for each group that would be synthetically feasible. We used the top-scoring groups, the original RPA small molecule papers [19,20,33], and the combinatorial library to design the first compound, DAZLN-55 (molecule **5**). The initial synthesis of the enone in molecule **3** was achieved through the aldol condensation of the aldehyde group on the 1-(6-methoxynaphthalen-2-yl)ethan-1-one with a methyl ketone. The enone was treated with hydrazine to achieve the 2-pyrazoline of molecule **4**. According to SwissADME [34], this compound has a logP of 3.97 and an estimated solubility log S of −5.33, a moderate solubility. The pyrazoline core was acylated with the cyclic anhydride, to add an oxopentanoic acid, the same oxoacid as used in TDRL-551, achieving compound **5**, DAZLN-55. The calculated physiochemical properties of DAZLN-55 are a logP of 3.91, and an estimated log S of −5.41. After testing this compound for affinity to hSSB1 (detailed below), we added a morpholino ethenone via an amide coupling, achieving molecule **6**, DAZLN-56. This has similar predicted physiochemical properties, with a logP of 3.20 and an estimated solubility of −4.73.

### 3.4. Dose–Response of DAZLN Compounds and TDRL-551

The three synthesized compounds, which included the intermediate DAZLN-51, were tested in comparison to TDRL-551 to determine if they could bind to hSSB1 using the AlphaLISA assay in triplicate. The *K_d_* of the compounds was determined using the AlphaLISA assay. EMSA dose–response experiments were performed to confirm the AlphaLISA results and demonstrate competition with ssDNA as an hSSB1 substrate.

DAZLN-51 showed no binding to hSSB1 in the AlphaLISA (Figure 3A, Table 2) or the EMSA format (Appendix A). The binding exhibited in the AlphaLISA experiment (Figure 3A) was suspected to be a result of interference with the assay. The DAZLN-51 EMSA is not included in Figure 3B because no inhibition was observed, but a gel is included in the Appendix A. DAZLN-56, on the other hand, exhibited a low level of binding in the AlphaLISA, but this was not reflected in the EMSA (Figure 3B, Table 2). DAZLN-55 and TDRL-551 exhibited binding to hSSB1 in both assays, although, based on the confidence interval, DAZLN-55 was less stable, with a much lower affinity than TDRL-551 (Figure 3, Table 2). The testing of DAZLN-55 and DAZLN-56 in the TruHits counter assay suggested that they interfered with the AlphaLISA assay above 10 μM, whereas TDRL-551 did not. The extension of DAZLN-56 with the morpholino ethenone had a negative effect on binding to hSSB1. While the addition of a morpholinopropane group has previously been shown to increase the potency of TDRL-551 binding to RPA [20], in the case of DAZLN-56, the addition of a morpholinoethanone group diminished the desirable physiochemical properties of the compound.

### 3.5. AlphaLISA Library Screen

As the DAZLN compounds demonstrated limited binding to hSSB1, we elected to conduct a screen of an 80-compound library obtained in an agreement with Merck KGaA, Darmstadt, Germany. The AlphaLISA screen discovered two hit compounds with an average inhibition above 45%, MS-ML24 and MS-ML25 (Figure 4). These hit compounds demonstrated distinctly higher values of inhibition. Conversely, compounds with calculated negative values are often those that interfere with the signal, exhibiting a larger response than the controls upon excitation. The hit compounds included glomerular epithelial protein 1 (GLEPP-1) inhibitors [35,36] and share very similar structures (Table 3). When we investigated the two highest hits, we found that one of the compounds (MS-ML26) was also a GLEPP-1 inhibitor, with a structure very close to the other two.

### 3.6. Dose–Response of Merck Compounds

An interrogation of the hit compounds from the screen determined that they had activities greater than that of TDRL-551 and DAZLN-55. Dose–response curves were determined for the three MS-ML compounds using an AlphaLISA assay and compared to a control In3-PS. Their affinity to hSSB1 was confirmed using EMSAs. All three compounds demonstrated the ability to prevent the binding of ssDNA to hSSB1. There is a distinct difference between the IC_50_ and K_d_ values of the EMSA and the AlphaLISA. This difference is likely a result of the optimal assay conditions in each case. For the EMSA to have a detectable signal, a different oligonucleotide probe-to-protein ratio is required to that in the AlphaLISA, which affects the amount of a compound that is required to displace the oligonucleotide. The two probe molecules are slightly different in each assay, with the AlphaLISA using a biotinylated oligonucleotide attached to a larger bead, while the EMSA uses a Cy5-labelled oligonucleotide. These assays make distinct differentiations in their binding strength.

MS-ML24 and MS-ML26 demonstrated comparable affinities to hSSB1 in the assays (Figure 5A,B, Table 3). The only difference between these two compounds being a fluorine in MS-ML26 instead of a hydroxyl group. There was an overlap of the IC_50_ range in the 95% confidence interval. MS-ML25 has a mildly weaker affinity to hSSB1. The only feature differentiating MS-ML24 from MS-ML25 is the ketone group between the central nitrogen atom and the 2-hydroxybenzoic acid, which allows for more conformational movement than the other two. All three compounds demonstrate increased potency compared to TDRL-551.

### 3.7. In Vitro Cytotoxicity

We tested the small molecules TDRL-551, DAZLN-55, DAZLN-56, and the MS-ML compounds using the CTG assay. TDRL-551 and the DAZLN compounds demonstrated no cytotoxic effect in U2OS cells following 48 h of incubation at concentrations spanning 30—0.0492 μM. The MS-ML compounds demonstrated cytotoxicity at concentrations above 2 μM (Figure 5C), with MS-ML-25 marginally showing the greatest effect. The original publication of these GLEPP-1 did not test inhibitors for their cytotoxic effect, but reported a lack of cytostatic activity in other monocytic human cell lines [35].

### 3.8. Cosolvent MD Simulations

Cosolvent MD simulations were performed to better understand the binding of the six compounds to hSSB1. This approach considers protein flexibility and allosteric binding, opening pockets that are not available in the apo form [37]. For each compound, we completed five MD replicas of 300 ns. After the simulations were complete, we used *k*-means clustering to determine the site that was most highly occupied by a compound across the five simulations, with the fraction of occupancy in each cluster presented in Appendix A. When a molecule occupied a site on the protein, this was defined as the binding site (BS1, BS2, etc.); see Appendix A.

Between the six compounds, the MS-ML scaffolds had the highest occupation of their respective top cluster (MS-ML24–52.1%; MS-ML25–59.9%; MS-ML26–81.9%), while TDRL-551 and the DAZLN compounds demonstrated lower stability (TDRL-551–44.2%; DAZLN-55–39.6%; DAZLN-56–43%). The residues interacting with the ligands in the representations of the top clusters are in Figure 6 (DAZLN-55, MS-ML24, and MS-ML25) and Appendix A. Appendix A depicts the interaction profile for each of the compounds in a 2D format. DAZLN-55 (Figure 6A), MS-ML24 (Figure 6B), and MS-ML25 (Figure 6C) bound to the ssDNA binding pocket, interacting with some of the aromatic residues—Tyr-74, Tyr-85, and Phe-78. π-π stacking with Tyr-85 and Phe-78 were observed in simulations of DAZLN-55 and MS-ML24, respectively. MS-ML26 also interacts via a π-π stacking interaction. Iin this case the interaction occurs with the Phe-98 on the interface that forms the protein–protein interaction with integrator complex subunit 3 (INTS3). From all these, DAZLN-55 and DAZLN-56 demonstrated the lowest stability.

### 3.9. Relative Binding Free Energy Calculations and Residue Decomposition

MM/GBSA calculations were conducted on the trajectories of the top clusters for each of the five novel compounds to understand the binding free energy calculation of protein–ligand complexes and the residues contributing to the energetics. MM/GBSA methods ensure alternate ranking based on the estimated binding free energy (ΔG_bind_) of a molecule, making it a useful tool in drug design research [38]. The relative binding free energy of the top clusters where the compound bound to hSSB1 revealed more negative free energies for the MS-ML compounds, reflecting the experimental results (Appendix A). The representations of these binding sites are presented in Figure 6 and Appendix A. The representations show that DAZLN-55, MS-ML24, and MS-ML25 bind to the ssDNA binding site, and the halogen-containing compounds bind to the opposite side of the protein. While DAZLN-55 has the lowest total free binding energy in the top binding site (Appendix A), the free binding energy of DAZLN-56 drops off in BS2 as the compound moves away from hSSB1. The per residue decomposition suggests that the most stable compounds are MS-ML24 and MS-ML25, which exhibited the best binding in EMSAs and were within the margin of error in the AlphaLISA binding assay (Table 3).

The MM/GBSA per residue decomposition analysis was used to provide insight into the interactions between the binding site and each of the compounds in the top cluster of each compound (Figure 7). This shows key residues that contribute to the binding of each compound. Residues with energy contributions more negative than −1.00 kcal/mol are presented in Table 4. The overall energies contributed by each residue are less favourable in DAZLN-55 than in MS-ML24 and MS-ML25. This supports the experimental data, which found that DAZLN-55 had a lower affinity. The decomposition of the pairwise energies is shown in Appendix A. DAZLN-55, MS-ML24, and MS-ML25 interact to varying degrees with the key aromatic residues Trp-55, Tyr-74, Phe-78, and Tyr85. MS-ML24 is the only one of these that interacts with Trp-55 above −1 kcal/mol. MS-ML25 demonstrates the strongest interactions with residues in the ssDNA binding site according to the total free energies (Appendix A) and the pairwise decomposition (Figure 7), suggesting that it has the highest affinity to ssDNA binding sites. Cys-99, a residue that is essential to hSSB1 oligomerisation [39], contributes free energy more negatively than −0.5 kcal/mol to DAZLN-56. Despite the similarity in free energy patterns to MS-ML24 and MS-ML25, DAZLN-55 shows a lower experimental affinity, a lower stability in the pocket based on the clustering (Appendix A), and lower free binding energies (Appendix A).

DAZLN-56 and MS-ML26 bound to the opposite face of hSSB1. The pairwise decomposition of these compounds (Appendix A) suggest a lower contribution of polar and electrostatic energies compared to the other compounds. These two compounds also interact with fewer residues in a different region of the protein, with a lower total binding free energy, for which electrostatic and polar solvation energy barely contribute to the affinity or are even detrimental to the binding. For those compounds that bind in the ssDNA binding site of hSSB1 (Appendix A), there are greater contributions from electrostatic and polar interactions.

## 4. Discussion

Targeting hSSB1 with a small-molecule inhibitor that binds to the OB-fold represents a promising mode of hSSB1 inhibition. Here, we describe the rational computer-aided design of compounds that bind to hSSB1. These molecules demonstrated a low affinity for hSSB1. A physical compound library was subsequently screened, revealing compounds with a high affinity to hSSB1 both at and below concentrations of 400 nM. We have demonstrated, for the first time, that the interaction between ssDNA and hSSB1 can be interrupted by small molecules, more specifically, by the MS-ML compounds that are described in this manuscript.

hSSB1 functions in DDR by binding to exposed ssDNA substrates. We employed the screening of MS-ML compounds to disrupt the binding interaction between hSSB1 and ssDNA. The competition format of the AlphaLISA and EMSA suggest that the compounds with an affinity to hSSB1 displaced the ssDNA from its binding site. According to the AlphaLISA binding assays, the most effective compounds were MS-ML24 (143 nM) and MS-ML26 (198 nM). These experimental methods cannot be used to determine where the compounds bind to purified hSSB1 and how they disrupt the binding of ssDNA. Cosolvent MD simulations were used to develop a stronger hypothesis regarding where the compounds bind.

Experimentally, the MS-ML compounds demonstrated a significantly higher affinity to hSSB1 than TDRL-551 or the DAZLN compounds. According to our simulations, the MS-ML compounds formed interactions with several residues in the respective binding pockets than the other compounds assessed. It appears that the essential interaction is the ability of the aromatic rings to form π-based interactions with the aromatic residues in the hSSB1 binding pocket. Traditionally, electrostatic interactions were thought to be the most important interactions when designing protein agonists. These findings suggest that the compounds’ capacity to form stable π interactions with protein residues is equally critical [40]. The greater steric profile of the indole and the 2-amino-1-morholinoethanone functional groups added to the DAZLN compounds appear to create a steric profile too large for effective binding to hSSB1, introducing unfavourable interactions.

The top clusters of each compound across the five simulations give some insight into how they interact with the hSSB1 monomer, which was supported by the pairwise free energy decompositions. The top cluster of MS-ML24 binds to the ssDNA binding pocket directly. The Phe-78 aromatic stacks with the salicylic acid functional group, while Tyr-85 and Ser-53 form hydrogen bonds with the salicylic acid, stabilizing the compound. Further, hydrophobic interactions occur between Thr-30, Thr-32, and Lys-33 and the diphenylethyne functional group. DAZLN-55 had similar interactions, with Phe-78 stacking with the indole and the methoxynapthalene. Tyr-85 further stabilizes the molecule in the pocket, while Ser-50 and Ser-73 form a donor and acceptor H-bond, respectively, to further stabilize the compound on the ssDNA binding surface. MS-ML25 sits in a similar orientation to MS-ML24, interacting with the key residues Tyr-85 and Arg-88, but in this case, Tyr-73 interacts with the salicylic acid of the molecule. The pairwise decomposition indicates that the basic residue Arg-88 plays an important role in binding DAZLN-55, MS-ML24, and MS-ML25, with one of the most negative contributions to binding these compounds. The rigidity of the methoxynapthalene causes it to extend out in a similar manner, but the amide group between the central nitrogen and the salicylic acid reduces the affinity. The carbonyl oxygen of the molecules, behaving as a H-bond acceptor, likely affects the binding negatively [41]. The fluorine in the MS-ML26 appears to cause the compound to bind to hSSB1 slightly outside of the ssDNA binding site in the top cluster. TDRL-551 did not bind to the ssDNA binding pocket in the cosolvent MD as it was predicted in the docking. This is because docking is a guided and biased method where the binding site is defined. The simulations indicated that the primary binding site occupied by TDRL-551 and DAZLN-56 is the interface that hSSB1 uses to interact with INTS3. INTS3 plays a role in controlling the hSSB1-mediated DDR [42]. In both cases, neither the fluorine in MS-ML26 or the iodine in TDRL-551 show interactions occurring in the simulations, where they are usually introduced to molecules for their unique binding properties [43]. An investigation as to whether these two compounds interfere in the protein–protein interaction between hSSB1 and INTS3 would provide insight as to whether these compounds are indeed binding at the interface predicted in the MD simulations. Crystallographic, NMR, or mutagenic studies are required to confirm the true binding sites of these compounds.

The AlphaLISA and AlphaScreen technologies have been successfully used in other high-throughput screening protocols [21]. We used the same technology on a smaller scale of 80 compounds, with a 2.5% hit rate from the initial screen. This initially missed MS-ML26, as the results on the plate were slightly lower than the threshold set for the screen. While there are a range of possible factors that contributed to the lower signals in the screen, small temperature changes and plate effects are known to impact signals [44].

Cosolvent MD simulations are an unbiased method used to detect binding hotspots, including cryptic binding sites [45]. As we have demonstrated, this is an unbiased technique that can also be used to predict the binding site of compounds on a protein to support experimental results. In this study, we see that the top binding compounds, MS-ML24 and MS-ML26, are also the compounds with the highest absolute value for their binding free energies, confirming the results from our two experimental protocols. DAZLN-55 and DAZLN-56 demonstrated a lower affinity to hSSB1, both experimentally and in the cosolvent simulations. According to the pairwise decomposition, there are fewer residues involved in the binding of the DAZLN compounds, and they contribute less to the binding free energy compared to the MS-ML compounds.

In the first phase of our study, we aimed to create new hSSB1 inhibitors by examining fragment libraries with substituted R groups on a core scaffold. Although this approach provided valuable insights into the characteristics of the binding pocket, we encountered several limitations with this technology. The compounds generated using such methodologies are typically extremely challenging to synthesize and, in some cases, impossible. Potential synthetic routes need to be determined, or else compounds with bioisostere R groups could be substituted to reach a synthetically feasible endpoint. Even if these changes are made, as we have demonstrated, the compounds do not necessarily bind as expected. Traditional fragment-based drug design (FBDD) uses in vitro methods such as surface plasmon resonance, isothermal calorimetry, nuclear magnetic resonance spectroscopy, or differential scanning fluorimetry to gradually grow the compounds into the target site of the protein [46]. Until progress is made in AI-assisted fragment hopping, in silico fragment growth needs to be supplemented with fragment binding experiments to better understand if fragments are interacting with the target protein, and how.

The compounds identified in this study that demonstrated the highest affinity to hSSB1 present some problems for further development. MS-ML24, MS-ML25, and MS-ML26 have all demonstrated chemotaxis in monocytic cell lines besides U2OS [35]. This study found that they were cytotoxic in the U2OS cell line, but the underlying mechanism causing cell death is not yet clear. Studies are necessary to determine if the compounds specifically inhibit hSSB1 repair activity or if they preferentially inhibit the activity of GLEPP-1 and other OB-fold proteins. Modifications to these compounds are also required to navigate beyond the existing patent [36], while simultaneously enhancing the affinity to hSSB1. Understanding the interaction of these compounds in a cellular environment to ascertain whether these molecules inhibit hSSB1’s functional role in DDR will give further insight to the selectivity of the compounds. Targeting the interaction between hSSB1 and ssDNA is predicted to have beneficial effects by reducing the acquisition of resistance to current cancer therapies. Without a well-designed and targeted carrier, this may have a detrimental effect on the DDR in healthy cells, potentially leading to negative side effects.

Here, for the first time, we have identified compounds that have a nanomolar affinity to the OB-fold protein hSSB1. The similarity of the ssDNA binding faces in OB-folds suggest that these compounds would be promiscuous towards other OB-fold proteins. Therefore, drug design campaigns that target OB-fold proteins need extensive testing for specificity.

## 5. Conclusions

We have identified two classes of small-molecule inhibitors that show the in vitro inhibition of the hSSB1–DNA interaction. In the last eight years, inhibitors of RPA have undergone iterative development. Based on the structural similarities between RPA and hSSB1, we hypothesized that RPA inhibitors might similarly bind to hSSB1. Using these compounds as a starting point, we designed and synthesized two novel compounds that were tested for their binding properties to hSSB1. While they did not have a strong affinity for hSSB1, a subsequent library screen using AlphaLISA technology revealed three compounds, with minor functional group differences, with a strong affinity to hSSB1. Cosolvent MD simulations predicted how these compounds bound to hSSB1, and the binding free energy studies for the three compounds from Merck kGaA, Darmstadt, Germany, aligned with the experimental data. These compounds have some structural similarities to the DAZLN compounds and TDRL-551, but have a central nitrogen atom instead of a pyrazole ring and more flexible R groups. These are the first small molecules that have been demonstrated to have an affinity to hSSB1 and are a further step towards inhibiting the DDR mechanisms in cells. These molecules require optimization to improve their binding affinities, and could later be developed in combination with chemo- or radiotherapy and, indeed, immunotherapy, to enhance the activity and prevent resistance to these therapeutic approaches when treating cancer. As SSB proteins occur not only in eukaryotic cells but also in prokaryotes, these scaffolds could be used to develop antibacterial therapeutics, adding to the knowledge base of scaffolds that interrupt the interaction between ssDNA and SSB proteins.

## Figures and Tables

**Figure 1 biology-12-01405-f001:**
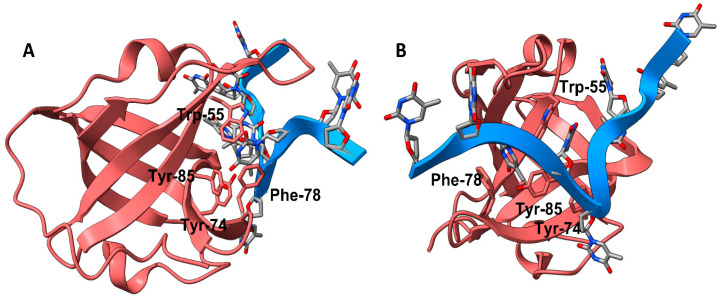
The binding of a poly-deoxythymidine (poly-dT, blue ribbon) strand to hSSB1 (coral ribbon), from the crystal structure of 4OWX of the SOSS1 complex. (**B**) is a 90-degree lateral rotation from (**A**). The key aromatic residues of the ssDNA binding pocket that interact with the strand are labelled accordingly.

**Figure 2 biology-12-01405-f002:**
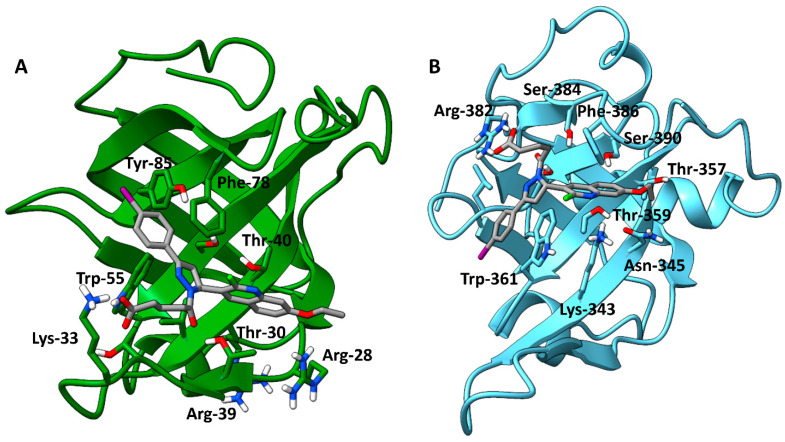
Illustration of the top-scoring pose of TDRL-551 when docked with hSSB1 (**A**) (PDB: 4OWX) and the DBD-B of RPA (**B**) (PDB: 1FGU). The images were rendered using ChimeraX version 1.4. Three of the four key aromatic residues in hSSB1 were predicted to interact with the compound, and in RPA, both aromatic residues were involved.

**Figure 3 biology-12-01405-f003:**
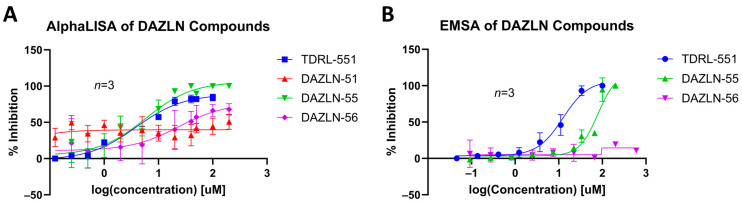
The binding of DAZLN compounds and TDRL-551 to hSSB1 using (**A**) AlphaLISA and (**B**) EMSA (*n* = 3). Compounds compete against the binding of In3-PD, an ssDNA oligomer that is used as a probe in the EMSA and bead binding partner in the AlphaLISA. Representative EMSA gels are included in Appendix A.

**Figure 4 biology-12-01405-f004:**
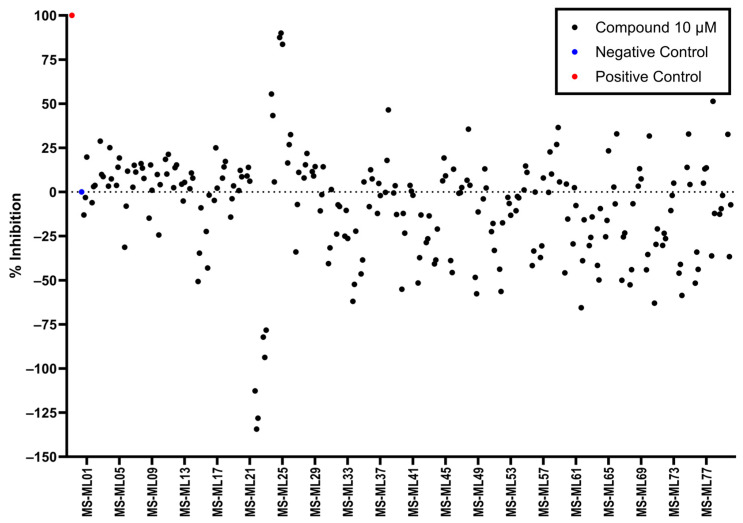
Scatter plot demonstrating the inhibition percent of each compound in the screen. Each of the compounds were screened in triplicate. The inhibition was calculated based on the positive and negative control signals, hence their locations at 0 and 100%. The compounds on the *x*-axis are labelled from MS-ML1 to MS-ML80. The positive and negative controls are on the far left.

**Figure 5 biology-12-01405-f005:**
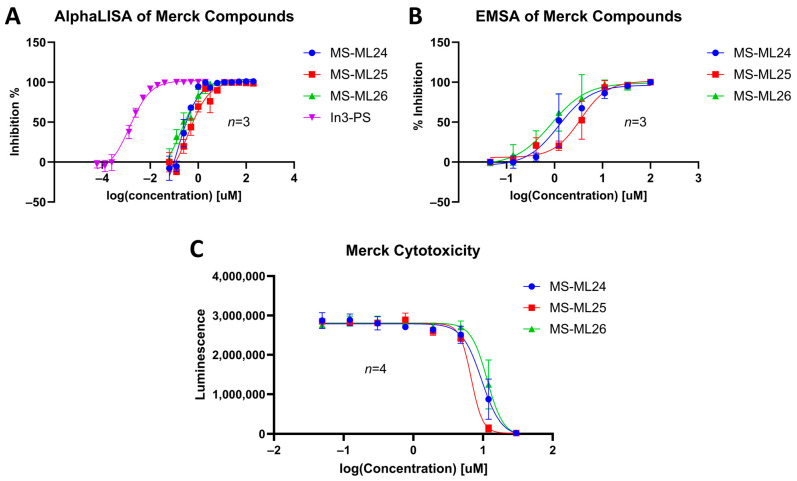
Comparing the binding of the MS-ML compounds to hSSB1 in two different assay formats—(**A**) using AlphaLISA and (**B**) EMSA. The known binder In3-PS, a thiolated oligonucleotide was used as a reference. In these cases, the compounds are out-competing In3-PD, an ssDNA oligomer that is used as a probe in the EMSA and bead binding partner in the AlphaLISA. (**C**) The cytotoxicity of the three MS-ML compounds over 48 h, tested on U2OS cells. Representative EMSA gels are included in Appendix A. For the dose–response assays, *n* = 3; for the cytotoxicity, *n* = 4.

**Figure 6 biology-12-01405-f006:**
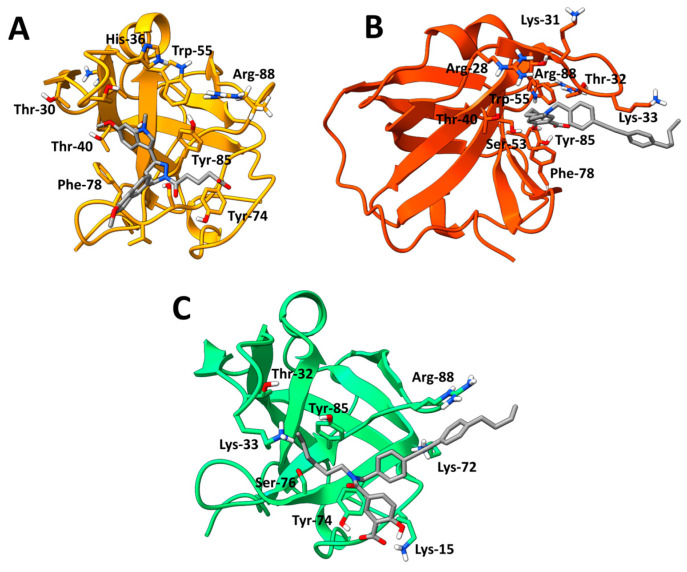
The representative position of the top clusters, where (**A**) DAZLN-55 (yellow), (**B**) MS-ML24 (orange), and (**C**) MS-ML25 (green) were bound to the ssDNA binding site of hSSB1 OB-fold. Key residues that the molecule interact with are labelled, based on the pairwise total free energy of the interaction. Other representative binding sites are available in the Appendix A. This figure was rendered using Chimera X [26,27].

**Figure 7 biology-12-01405-f007:**
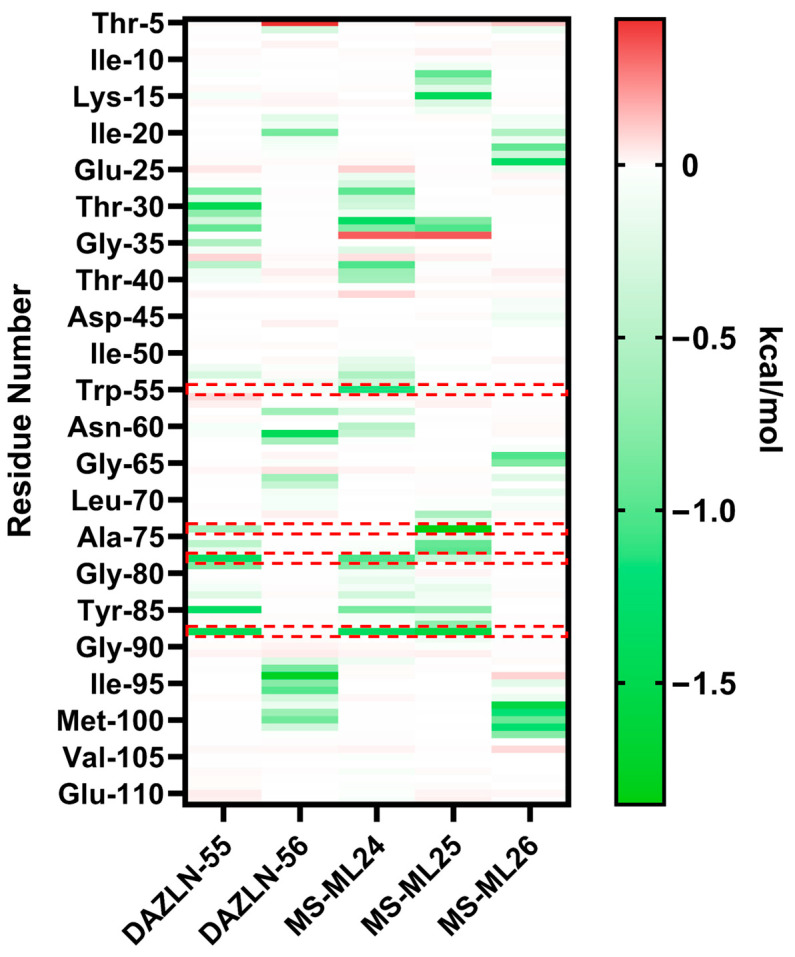
Heatmap of the hSSB1 residues interacting with the novel compounds. The pairwise total free energy of the residues in the protein complex in relation to the compound. These are calculated from the trajectory of the top cluster across five runs. The key residues in the ssDNA binding site, Trp-55, Tyr-74, Phe-78, and Tyr-74, are highlighted with the dashed red boxes.

**Table 2 biology-12-01405-t002:** The synthesized DAZLN compounds and TDRL-551, including their binding affinity in the AlphaLISA and EMSA binding assays. The compound reference numbers are listed according to the synthesis methods. The K_d_ and IC50 values were calculated with GraphPad Prism 10.0.2. The 95% CI denotes the confidence interval calculated for the K_d_ and IC50 values.

Compound Name	Compound Reference	Structure	K_d_ (μM) AlphaLISA	IC50 (μM) EMSA
TDRL-551		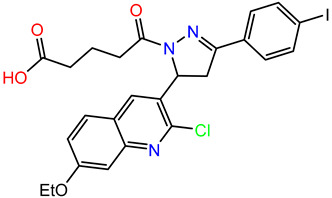	3.888 95% CI 2.503 to 5.859	12.48 95% CI 9.325 to 18.38
DAZLN-51	**4**	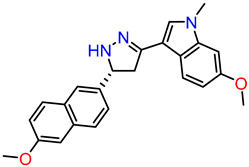	<100 μM	N/A
DAZLN-55	**5**	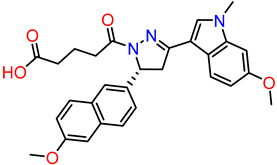	5.793 95% CI 3.188 to 10.31	79.56 95% CI 60.44–751.9
DAZLN-56	**6**	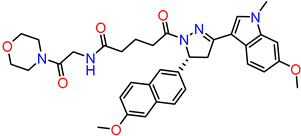	23.95 95% CI 5.738–88.26	N/A

**Table 3 biology-12-01405-t003:** The three hit compounds from the small molecule library screen of Merck KGaA, Darmstadt, Germany. Listed next to each compound is the Kd determined in the AlphaLISA assay; the IC50 from the EMSA analysis; and the IC50 from the cytotoxicity assay. The Kd and IC50 values were calculated with GraphPad Prism 10.0.2. The 95% CI denotes the confidence interval calculated for the Kd and IC50 values.

Compound	Structure	K_d_ (μM) AlphaLISA	IC50 (μM) EMSA	IC50 (μM) Cytotoxicity
MS-ML24	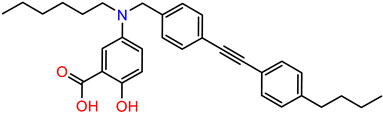	0.143 95% CI 0.0948 to 0.206	1.238 95% CI 0.629 to 2.76	9.558 95% CI 8.21 to 11.1
MS-ML25	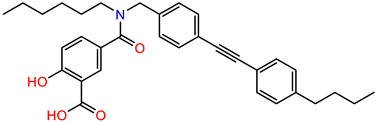	0.391 95% CI 0.274 to 0.555	3.504 95% CI 2.30 to 5.07	6.78 95% CI unbound to 7.47
MS-ML26	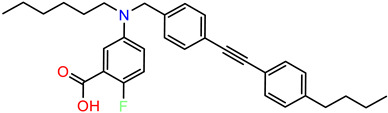	0.198 95% CI 0.1273 to 0.2969	0.857 95% CI 0.239 to 1.60	11.45 95% CI 10.1 to 13.9

**Table 4 biology-12-01405-t004:** The residues with a pairwise binding free energy below a cut-off of −1.00 kcal/mol based on MMGBSA calculations on the frames from the top cluster of each compound trajectory.

DAZLN-55		DAZLN-56		MS-ML24		MS-ML25		MS-ML26	
Thr-30	−1.48 ± 1.98	Lys-94	−1.76 ± 1.74	Arg-88	−1.38 ± 2.33	Tyr-74	−1.85 ± 1.32	Phe-98	−1.59 ± 0.984
Arg-88	−1.46 ± 2.51	Leu-61	−1.44 ± 1.65	Thr-31	−1.35 ± 1.37	Arg-88	−1.61 ± 1.84	Leu-21	−1.26 ± 0.928
Phe-78	−1.37 ± 1.05			Trp-55	−1.14 ± 1.59	Lys-15	−1.43 ± 1.39	Val-101	−1.26 ± 0.918
Tyr-85	−1.36 ± 1.54			Phe-78	−1.03 ± 0.857	Lys-33	−1.01 ± 1.72	Cys-99	−1.21 ± 0.904
				Val-38	−1.01 ± 0.708			Pro-64	−1.03 ± 0.726

## Data Availability

Data available in a publicly accessible repository that does not issue DOIs: https://github.com/zShores/hSSB1_small_mol_trajectories.

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
