# Peer review of "An Exploration of Small Molecules That Bind Human Single-Stranded DNA Binding Protein 1"

_biology, 2023, doi:10.3390/biology12111405_

Round 1

Reviewer 1 Report (Previous Reviewer 2)

Comments and Suggestions for Authors

The revised manuscript satisfactorily addressed my concerns in my previous review. However, the modulation of repair activity (gammaH2Ax foci assay) by the potential hSSB1 inhibitors is inconclusive. The new results presented in the revised paper are too preliminary. The authors should take Fig. 6 out of the paper. They also did not include the DNA repair assay in the methods section. However, all the binding results are fine and corroborate their overall conclusions and the paper's title.  

Author Response

Thank you for feedback on the revised version of the paper. We have followed your advice and removed the preliminary data presented in Figure 6 and the relevant methods and discussion. The DNA repair assay was the immunofluorescence experiment which has now been removed.

Reviewer 2 Report (Previous Reviewer 1)

Comments and Suggestions for Authors

While the manuscript is improved, there still remains significant concerns.

Figure legend 1 and line 81 of page 3:  Reference to the DNA as Thymine (polyT) is inaccurate.  It should be thymidine or deoxythymidine or deoxyribosylthymine and poly dT. 

 Page 5 the chemical structure is also labeled as 1 and is the second structure and need to be labeled 2. 

 Line 339,  505 is not a derivative of 551.  In fact 551 is a derivative of 505.  505 was published first Shuck et. al. and 551 published later Mishra et al.

Line 342.  The reference to compound 2 I assume is for the second compound listed a 1. 

Figure 6 page 18:  The IF figures are not adequate to interprate.  There are no apparent foci detectable at this magnification and it is unclear what is then being quantified. 

I am unclear what it means when you refer to the compounds as “having H2AX activity”.  The description of these experiments is scientifically inaccurate and inadequate for publication.   

More robust statistics are needed as it is not clear to which values are being compared in the single statement referring to ANOVA test.  Is there really a difference in the 1 hours time point in the individually treated cells versus DMSO?  The 24 hour data is potentialy interesting as ML26 and ML25 seem to repair more than the DMSO as indicated by fewer foci. 

There is no explanation of the ML24 image which appears to have fewer cells. 

This ML26 H2AX data however is inconsistent with the cytotoxicity data which shown the highest IC50, yet has fewer H2AX foci at 24 hours.  This should be addressed. 

Unfortunately the new H2AX experiment leads to more questions than answers and does not help justify the conclusions that any of the cytotoxicity data are in fact a function of HSSB1 inhibition. 

Referring to this as a pilot study does not abdicate ones responsibility to accurately described the data.  I would recommend the complete analysis to confirm an impact on DNA damage and repair.  Also, g-H2AX is simply a surrogate for damage and needs to be indicated as such.

Comments on the Quality of English Language

There remains many errors in grammar and a general alck of clarity in the presentation.

Author Response

Thank you for reviewing the updated version of the paper. We are grateful for the time you spent doing so and the insightful comments that you have made. Please see the following responses.

We have rectified the usage of thymine throughout the paper to deoxythymidine as was used in the original paper. (Figure 1, page 2; lines 76-77, page 3)

This was a formatting error between tracked changes and submission to the MDPI portal. This has been fixed. (Compound 2, page 4)

We have rectified the erroneous use of language, having realised that TDRL-551 was a derivative of TDRL-505. (line 328, page 10)

Based on a comment from another reviewer, and extensive discussion amongst the authors, we have decided to remove this figure and rephrased around the binding of the MS-ML compounds to hSSB1. The IF data was pre-emptive and we thought it might give some further insight to the results, but upon further examination, it made it less clear. Further work needs to be done to comprehensively investigate if the MS-ML compounds directly bind to other OB-fold proteins or if they affect the DNA repair mechanisms of cells. The focus of this paper was to present the screen and computational work on hSSB1 as a druggable target, with dose-response data for the compounds with purified protein.

Round 2

Reviewer 2 Report (Previous Reviewer 1)

Comments and Suggestions for Authors

The revised manuscript is improved and presents novel interesting data.

This manuscript is a resubmission of an earlier submission. The following is a list of the peer review reports and author responses from that submission.

Round 1

Reviewer 1 Report

Comments and Suggestions for Authors

Comments on the Quality of English Language

Author Response

Dear Reviewer 1,

We are immensely grateful for the constructive feedback you gave on the paper. It raised some key issues with the paper that we had overlooked, and we think have our responses to your comments have greatly improved the paper. We have attached our responses with references to where they have been fixed in the manuscript, and look forward to a positive response.

Kind Regards,

Mr Schuurs

Reviewer 2 Report

Comments and Suggestions for Authors

Introduction:

Line 45: Repair activity should reduce, not introduce mutations; the statement seems contradictory.

Line 79: Cantered>>typo

Line 83 and in many other places throughout the manuscript: references were missing.

Approach/Methodology:

The authors used two approaches: virtual and biochemical, for screening compounds for hSSB1 binding. All methodologies are appropriate for this study

Results:

Binding assay and Molecular dynamics simulations results are acceptable and clearly show that Merck compounds are potential binders of hSSB1. However, the approaches are either indirect or virtual. Some assays, such as the Biacore experiment with immobilized hSSB1 to monitor direct binding of the small molecules, should be tried to see if these small molecules bind directly to the protein but not the DNA. They also should try some in vitro or in-cell DNA repair activity assay (functional assay) involving hSSB1 to test the efficacy of these molecules as inhibitors for hSSB1’s repair activity. This way, the cytotoxicity assay done here has no meaning. The specificity is not clear. It could be GLEPP-1 (not hSSB1), which is targeted in the cells and that caused cell death. They should combine these Merck compounds with some DNA damaging agent to determine whether these compounds inhibit specifically repair activity of hSSB1 in cells and modulate cell’s sensitivity to DNA damaging agents. 

Comments on the Quality of English Language

Minor editing is required.

Author Response

Dear Reviewer 2,

We are immensely grateful for the constructive feedback you gave on the paper. It raised some key issues with the paper that we had overlooked, and we think have our responses to your comments have greatly improved the paper. We have attached our responses with references to where they have been fixed in the manuscript, and look forward to a positive response.

Kind Regards,

Mr Schuurs
